# Characterization of Antimicrobial Composite Edible Film Formulated from Fermented Cheese Whey and Cassava Peel Starch

**DOI:** 10.3390/membranes12060636

**Published:** 2022-06-20

**Authors:** Gemilang Lara Utama, Isfari Dinika, Siti Nurmilah, Nanang Masruchin, Bambang Nurhadi, Roostita Lobo Balia

**Affiliations:** 1Faculty of Agro-Industrial Technology, Universitas Padjadjaran, Jalan Raya Bandung-Sumedang Kilometer 21, Jatinangor 45363, Indonesia; isfaridinika@gmail.com (I.D.); sitinurmilah23@gmail.com (S.N.); bambang.nurhadi@unpad.ac.id (B.N.); 2Center for Environment and Sustainability Science, Universitas Padjadjaran, Jalan Sekeloa Selatan I No. 1, Bandung 40134, Indonesia; 3Research Center for Biomass and Bioproducts, National Research and Innovation Agency of Indonesia (BRIN), Cibinong 16911, Indonesia; masruchin@biomaterial.lipi.go.id; 4Research Collaboration Center for Biomass and Biorefinery between BRIN and Universitas Padjadjaran, Jatinangor 45363, Indonesia; 5Veterinary Study Program, Faculty of Medicine, Universitas Padjadjaran, Jalan Raya Bandung-Sumedang Kilometer 21, Jatinangor 45363, Indonesia; roostita.balia@unpad.ac.id

**Keywords:** antimicrobial, cassava peel, edible film, fermentation, starch, whey

## Abstract

Antimicrobial composite edible film can be a solution for environmentally friendly food packaging, which can be made from fermented cheese whey containing an antimicrobial agent and cassava peel waste that contains starch. The research aims to determine the formulation of fermented cheese whey and cassava peel waste starch, resulting in an antimicrobial composite edible film with the best physical, mechanical, and water vapour permeability (WVP) properties, as well as with high antimicrobial activity. This research was conducted using experimental methods with nine composite edible film formulation treatments with three replications. Three variations in the fermented cheese whey and cassava peel starch ratio (*v*/*v*) (1:3, 1:1, 3:1) were combined with variations in the addition of glycerol (20%, 33%, 45%) (*w*/*w*) in the production of the composite edible film. Then, the physical characteristics such as elongation at break, tensile strength, WVP, colour, and antimicrobial effect of its film-forming solution were observed. The results showed that 24 h of whey fermentation with *Candida tropicalis* resulted in an 18.50 mm inhibition zone towards *Pseudomonas aeruginosa*. The best characteristic of the film was obtained from the formulation of a whey:starch ratio of 1:3 and 33% glycerol, which resulted in a thickness value of 0.21 mm, elongation at break of 19.62%, tensile strength of 0.81 N/mm^2^, WVP of 3.41 × 10^−10^·g/m·s·Pa at a relative humidity (RH) of 100%–35%, and WVP of 9.84 × 10^−10^·g/m·s·Pa at a RH of 75%–35%, with an antimicrobial activity towards *P. aeruginosa* of 5.11 mm.

## 1. Introduction

Two food products, cheese and cassava, have become major concerns regarding their by-products resulting from the production processes. The high cheese whey volume resulted in up to 90% in every cheese production cycle [1,2]. Meanwhile, cassava production in Indonesia reached 21.80 million tons per year, resulting in 15–20% cassava peels or up to 4.7 million tons per year [3,4]. By-products such as cheese whey and cassava peels can pollute the environment when directly disposed. Cheese whey has a Biochemical Oxygen Demand (BOD) above 35,000 ppm and Chemical Oxygen Demand (COD) above 60,000 ppm [5].

Meanwhile, cassava peels could contain hazardous hydrogen cyanide (HCN) in cyanogenic glucose, which is toxic to human health [6,7]. However, cheese whey and cassava peels can be utilized to reduce the potential of environmental pollution as the solution to the problem of agro-industrial wastes.

Based on nutritional aspects, cheese whey still has many benefits. Whey contains 55% of the total nutrients contained in milk, which includes 6.3% of total solids that consist of lactose (4.9%), protein (0.7%), fat (0.1%), and ash (0.5%) [8,9]. In addition, whey also has some functional properties derived from amino acids and peptides. Peptides in whey have functional properties such as antioxidant, antimicrobial, anti-hypertensive, anti-cancer, opioid, and immunomodulatory functions [10]. Concerning the potential and functional properties, the presence of amino acids and peptides in whey can be enhanced by fermentation [11].

Cassava peels contain starch that can be a raw material for making an edible film with polysaccharides based on good mechanical characteristics. Cassava peels are 75% (*w*/*w*) starch, 62.51% of which is amylopectin and 21.70% is amylose [12,13]. The amylopectin: amylose ratio, approaching 70:30, makes it a good film-forming material [14,15]. Due to the polysaccharide properties soluble in water, the polysaccharide-based film formulation has high water vapour permeability (WVP). Some research also found that combining polysaccharides with protein-based materials in composite film formulation will enhance the WVP properties. The solubility of the film decreases as the concentration of cassava starch increases, whereas the dominant whey proportion is more resistant to thermal decomposition [16]. Reformulation polysaccharides with protein could increase hydrophobic characteristics that resulting in barrier properties towards water vapour, oxygen, and lipids [17].

The utilization of cheese whey and cassava peel starch into a composite edible film is expected to reduce agro-industrial waste pollution and plastic packaging [16,18,19]. As the second-largest plastic waste producer globally, Indonesia produced 3.2 billion kg/year of plastic waste after China (8.8 billion kg/year) [20]. Plastic is the typical material used in food packaging, due to its good optical and physical barriers [21]. However, plastic cannot be naturally degraded. Plastic is a single-use non-biodegradable packaging material that is disposed of, emitting greenhouse gases and posing potential environmental concerns to human health [22]. Therefore, plastic packaging needs to be replaced with biodegradable alternatives. In that case, cheese whey and cassava peel starch have potential to develop as biodegradable packaging that can contribute to resolving the problems of plastic packaging while also reducing the potential of cheese and chip waste problems. Film biodegradation can occur due to aerobic or non-aerobic fermentation of microorganisms that secrete extracellular enzymes, hydrolysing polymer chains in packaging materials, producing reduced molecular weight degradation products, and forming environmentally safe product metabolites [23].

In its application, the manufacture of composite edible films using protein bases has been widely carried out. Of all protein sources, soy protein and whey protein are the most studied components because they are easy to obtain [24,25,26,27]. Furthermore, in the manufacture of composite edible film, proteins, polysaccharides, and fats can be added. The addition of soybean protein to starch-based edible film composites resulted in a lower water solubility value than the whey protein composite; this indicates that the whey protein composite film showed a better reaction to water vapour [28]. Composite edible film from whey protein and starch showed superior characteristics, with thinner film thickness, lower tensile strength, and higher flexibility than soy protein [24,29].

In addition to that, cheese whey can have functional effects and antimicrobial properties against various bacteria and yeasts if further processed through fermentation. Lactoferrin was found to be a dominant peptide in whey that showed potential as a functional compound, with a concentration of 1.5 g/L [8]. Through fermentation, lactoferrin can be hydrolysed by the pepsin enzyme to release lactoferricin, which has broad-spectrum antimicrobial properties, so that the fermented whey can show a significant antimicrobial effect [30]. Lactoferrin will attack lipopolysaccharides in Gram-negative bacteria and lipoteichoic and teichoic acids in Gram-positive bacteria; meanwhile, lactoferricin can interfere with and change the permeability of bacterial membranes so that macromolecular biosynthesis is inhibited and causes cell death [31,32].

In making a composite edible film, the characteristics are strongly affected by the precise formulation of material-based plasticizers. The optimal amount of plasticizer used is 20–45% (*w*/*w*), because the matrix will be brittle if it is too low and sticky if it is too high [33]. The formulation of starch–protein materials for edible film composites has been widely practiced. However, the use of agricultural wastes such as cassava peel mixed with fermented cheese whey to increase the physical, mechanical, and antimicrobial characteristics is rarely conducted. This research aimed to determine the best formulation to make the best film characteristics regarding the physical, mechanical, water vapour permeability, and antimicrobial properties.

## 2. Materials and Methods

Cheese whey was obtained from KPBS Pangalengan, Bandung District, Indonesia, and cassava peels were obtained from a small–medium cassava chips enterprise in Rancasalak, Desa Cimaung, Bandung District, Indonesia. Other materials were glycerol (P&G Chemical Asia, Selangor, Malaysia), NaCl (Merck, New Jersey, USA), Plate Count Agar (Oxoid, UK), Potato Dextrose Agar (Oxoid, Hampshire, UK), Yeast Mould Agar (Oxoid, Hampshire, UK), Nutrient Broth (Oxoid, Hampdshire, UK), chloramphenicol (Novapharin, Gresik, Indonesia), and *Candida tropicalis* as a starter that was isolated and purified from mozzarella cheese whey. The instruments that were used were an Incubator (Memmert, Schwabach, Germany), Pasteurizer (Agrowindo, Malang, Indonesia), Universal Testing Machine (Shimadzu AG-IS 50kN, Kyoto, Japan), spectrophotometer (CM-5 Konica Minolta, Tokyo, Japan), desiccator (Normax, Marinha Grande, Portugal), and an oven blower (Maksindo, Bogor, Indonesia).

This research was conducted using experimental methods with the formulation of the composite edible film with 9 treatments with 3 replications. There were 3 variation ratios of the fermented cheese whey and cassava peel starch (*v*/*v*) (1:3, 1:1, 3:1) and 3 variations in the plasticizer (glycerol) added (20%, 33%, 45%) (*w*/*w*) [24]. The observations were the physical characteristics such as elongation at break, tensile strength, WVP and colour, and the antimicrobial effect of its film-forming solution. The treatment and formulation are listed in Table 1 and Table 2.

### 2.1. Cassava Peel Starch Production

Starch from cassava peels was made by crushing the clean cassava inside the peel using a blender with the addition of aquadest in a ratio of 1:2 *v/v* cassava and aquadest. The process was followed by squeezing it with a filter cloth to obtain the extract and precipitate it for 12 h. Next, the supernatant was removed, the sediment washed with the addition of aquadest in a ratio of 1:1, and then it was precipitated for 1 h repeatedly until the supernatant became clear (2–3 times). The pure white sediment was then dried in an oven blower at 60 °C for 30 min of modification [4].

### 2.2. Composite Edible Film Production

Starch powder containing 70.43 ± 0.25% starch and 15.40 ± 0.01% water content was diluted in aquadest (5%) and heated at 85 °C for 15 min until the homogeny was optically clear. The temperature was lowered to 60 °C and then it was mixed with fermented whey (94.02 ± 0.06 moisture, 0.88% protein content) and glycerol. After another 15 min, the film-forming solution was cast and dried in an oven blower at 25 °C and RH 40% for two days. The film was peeled from the casting and placed in the desiccator at 25 °C and RH 40% for two days of modification [24].

### 2.3. Physical Characterization of Edible Film

The physical characteristics of the antimicrobial composite edible film are thickness using a micrometre screw, colour test with a spectrophotometer (CM-5 Konica Minolta), and elongation at break and tensile strength using the Universal texture Machine (Shimadzu AG-IS 50kN) [34].

### 2.4. Edible Film Antimicrobial Activity Identification

Antimicrobial activity testing was performed on fermented whey and the antimicrobial composite edible film produced against bacteria *Escherichia coli* (*E.coli*), *Salmonella thypimurium* (*S. thypimurium*), *Staphylococcus aureus* (*S. aureus*), and *Pseudomonas aeruginosa* (*P*. *aeruginosa*) using the diffusion well for the fermented whey and diffusion disc method for the film [35].

## 3. Results

### 3.1. Physical Characteristics of Antimicrobial Composite Edible Film

The results of the edible film appearance are shown in Figure 1. Good film characteristics were formed in the whey: starch ratio of 1:3 (WP13), also shown by all glycerol treatment variations (20%, 33%, and 45%), in which the film could be peeled from the casting, was flexible, and could be folded. The WP13 treatments were a transparent yellowish-white colour with a soft, matte, and misty surface. The backside of the film was greasy and glossy; it was the surface that had contact with the casting. Unlike WP13, the physical characteristics of the film with a larger whey amount ratio of 3:1 (WP31) had the worst result. It was the same as WP11: fragile, easily broken, and could not be folded. The WP31 was more yellowish than WP11, which showed a light yellowish-white colour.

Physical characteristics such as thickness, tensile strength, and elongation at break are shown in Table 3 and Figure 1 and Figure 2. The result showed that the best characteristics were of WP13B, with the best elongation at break value (19.62 ± 7.20%), the thinnest (0.21 ± 0.00 mm), and the second-largest tensile strength (0.81 ± 0.29 Nmm^2^) after WP13A.

Table 4 is the result of water vapour permeability (WVP). There are two types of WVP, RH100%–35% and 75%–35%, based on the difference in the solution used. Therefore, it can represent the optimum water content of food to be packaged with this composite edible film. The lowest WVP was held by WP13B (3.41 ± 1.13 × 10^−10^·g/m·s·Pa for RH 100%–35% and 9.84 ± 1.50 × 10^−10^·g/m·s·Pa for RH 75%–35%), which can be claimed as the best treatment based on the different RH applied. Even though the WP13A had the lowest glycerol addition of 20%, the standard deviation showed non-significant results compared to WP13B. The colour test observations obtained the value of L * a * b *, which can be seen in Table 5. Based on information on the colorhexa (https://www.colorhexa.com/, accessed on 16 May 2021), the results of the observations show a light greyish-orange colour.

### 3.2. Antimicrobial Characteristics of Antimicrobial Composite Edible Film

Antimicrobial characteristics were observed on unfermented whey, fermented whey after 24 h, and whey formulated into a film-forming solution by the best method. The test bacteria used were *Escherichia coli*, *Salmonella typhimurium*, *Staphylococcus aureus*, and *Pseudomonas aeruginosa*. The results of the observations can be seen in Table 6. The results showed antimicrobial activity towards spoilage bacteria, namely *P. aeruginosa*. The antimicrobial strength showed a clear zone of 18.50 mm in the sample of fermented whey using *C. tropicalis* for 24 h.

## 4. Discussion

### 4.1. Physical Characteristics of Antimicrobial Composite Edible Film

The results of the edible film appearances and physical characteristics are shown in Figure 1 and Figure 2 and Table 1, Table 2 and Table 3, respectively. Analysis of thickness, tensile strength, elongation at break, water vapour permeability, and colour was carried out. Elongation at break is affected by starch content. The higher starch composed of amylose will make the film more flexible and strong [36]. Otherwise, adding glycerol to starch-based edible films can also increase the resulting elongation. Glycerol molecules penetrate more easily between the starch polymer matrix and weaken the intermolecular forces between starch polymers because glycerol has a small molecular size, many hydroxyl groups, and high hydrophilic properties [37]. This results in the rigid film structure becoming softer and the elongation becoming higher, which causes the film to become more flexible and pliable [38].

A thin layer was expected to result in a 0.050–0.250 mm thickness. This thickness shows the ability to prevent the passage of water vapour, prevent the loss of volatile components, and increase the product’s appearance through colour improvements [39]. Furthermore, the thickness is essential to make sure the layer is flexible, could roll, could immerse, could give maximum contamination protection, and potentially enhance the nutritional value [40].

Average tensile strength < 1 N/mm^2^ was obtained on materials with the best characteristics. This formulation attempts to provide an alternate packaging for soft mozzarella cheese with relatively high water content. Based on Nordin et al. [41], the tensile strength of the resulting edible film decreases due to the increase in glycerol concentration. This is because glycerol can interfere with amylose in the starch matrix through hydrogen bonds, which causes the intramolecular attraction between starch chains to decrease and can increase the mobility of the polymer chains so that the resulting tensile strength decreases and the film becomes more flexible [42]. This composite’s tensile strength can also be improved by adding food-grade additives, which can be used more widely.

All of the materials formulated showed roles in determining the properties of the edible film that resulted. The cassava peel starch composed of amylose could form hydrogen bonds, especially when the heat applied could form a three-dimensional net that trapped the water and resulted in a strong gel [43,44]. The addition of cheese whey resulting in an increased amount of protein that also results not transparent or cloudy film colour and the increase of globular shaped protein that results bad matrix bond, which is lowers the physical and mechanical properties of the film [45,46].

Based on the colour test observations, the resulting edible film tended to be more yellow. The brightness of the cheese whey addition increased due to the yellowish pigments in the cheese whey in the form of riboflavin and carotene [47]. 

As seen in the results, the use of glycerol as a plasticizer needed to be noticed, as a high concentration of glycerol caused the increase in WVP because the glycerol is hydrophilic [48]. As the concentration of glycerol increases, the water vapour permeability of the edible film also produced increases [19]. This is because glycerol has relatively small hydrophilic molecules, reducing intermolecular attraction, increasing molecular mobility, and facilitating water vapour migration [49]. Glycerol is also hygroscopic, so the attraction to water molecules is high. This causes easier water diffusion [50]. Otherwise, the low glycerol content can cause the film matrix to become brittle and hollow and lower the mechanical properties so that the hydrophobic matrix from starch can absorb water easier [33]. The best treatment (WP13B) showed that the lowest cheese whey ratio with low glycerol addition results in the lowest WVP.

The acidity of the formula solution is close to the whey isoelectric pH of 5.9, which makes the protein coagulated, solidified, and difficult to disperse [51]. Low acidity keeps the protein denatured so that the sulfhydryl and hydrophobic bond can open and re-bond when the drying process occurs, which enhances the mechanical characteristics of edible films [44,52].

The edible film formulation of fermented cheese whey and cassava peel starch (1:3) with the addition of glycerol (33%) (WP13B) results in the best physical characteristics regarding the thickness and tensile strength elongation at break, water vapour permeability, and the colour. This proportion is in line with the formulation of whey:cassava 1.30:3.17% to produce mechanical and physicochemical properties: higher thickness (1.128 mm), higher tensile stress (1.92 N/mm^2^), higher elongation (40.4%), yet lower lightness (89.9) [18]. There is an effect of increasing the tensile strength of materials with high contents of cassava starch, but this can disrupt the continuity of the matrix when it is in high proportion. In comparison, another study showed that the 67.50:7.50% (9:1) whey:cassava starch formulation was a film with a higher thickness (0.70 ± 10 mm) [16]. The film with the highest whey proportion is more stable regarding water and thermal decomposition due to a protein having more hydrophobic characteristics to increase the barrier properties against water vapour, oxygen, and lipids [16,17]. The development of edible polymer blend films is a critical stage in developing packaging films. The compatibility of the edible film polymer mixtures for the product is essential in choosing the edible film formulation.

### 4.2. Antimicrobial Characteristics of Antimicrobial Composite Edible Film

Antimicrobial properties can be obtained by fermenting cheese whey with native yeast. Table 6 shows that *P. aeruginosa* could be inhibited by fermented cheese whey. However, after the fermented cheese whey was applied to the film-forming solution, the zone of inhibition decreased to 5.11 mm. The antimicrobial impact on the growth of *P. aeruginosa* depends on the concentration [53]. The decrease in antimicrobial activities indicates a decrease in an antimicrobial substance due to the formulation. The formulation resulting in lower ratio of fermented cheese whey on the film solution which resulted low antimicrobial activities on the edible film. 

The antibacterial activities did not show most of the pathogenic bacteria, such *S. aureus*, *S. thypimurium,* and *E. coli*. Following Adegbehingbe and Bello, [51] mentioned no bactericidal activity of the fermented whey against the pathogenic microorganisms. At the same time, some growth inhibition was found due to the activity of lactoperoxides and lactoferrin, which are present in whey. Lysozymes, lactoperoxidase, and lactoferrin were found in whey which could potentially inhibited the growth of pathogenic bacteria [54].

Antimicrobial characteristics can result from the fermentation of cheese whey with the addition of native *C. tropicalis,* which can produce antimicrobial peptides [1]. The peptides with low molecular weight showed higher antibacterial activities towards *P. aeruginosa*, while the small peptides did not inhibit the *S. aureus* growth [55]. The anionic peptides showed good antimicrobial activities only toward *P. aeruginosa* and *Pseudomonas* sp. There was no inhibition of good antimicrobial activities towards *P. aeruginosa* and *Pseudomonas* sp., and there was no inhibition detected towards *E. coli* and *S. aureus* [53]. The inhibition activities of *P. aeruginosa* also result from the acid environment generated by fermentation [56]. The other pathogenic bacteria, such as *S. thypimurium*, *E.coli*, and *S. aureus*, showed survival ability, growing and even multiplying in an acidic environment during the fermentation of some milk products [57].

## 5. Conclusions

Whey fermentation with *C. tropicalis* resulted in an inhibition zone of 18.50 mm for *P. aeruginosa*. The antimicrobial composite edible film formulation showed the best characteristics in the WP13B formulation with a thickness value of 0.21 mm, an elongation at break value of 19.62%, tensile strength of 0.81 N/mm^2^, WVP of 3.41 × 10^−10^ g/msPa for RH 100%–35%, and 9.84 × 10^−10^ g/msPa for RH 75%–35%. The antimicrobial activity for *P. aeruginosa* was 5.11 mm. The composite edible film can be produced with fermented whey and starch to gain good mechanical characteristics and a good barrier to prolong the shelf-life of food, especially vegetables. Furthermore, fermented cheese whey as an edible film material can form an active packaging system. This packaging system can be applied as the best film for continued future work for cheeses, fresh-cut fruits, or high or potentially high *P. aeruginosa*-contaminated vegetables. Future studies should be carried out to evaluate thermal stability, water absorption test, and biodegradability test.

## Figures and Tables

**Figure 1 membranes-12-00636-f001:**
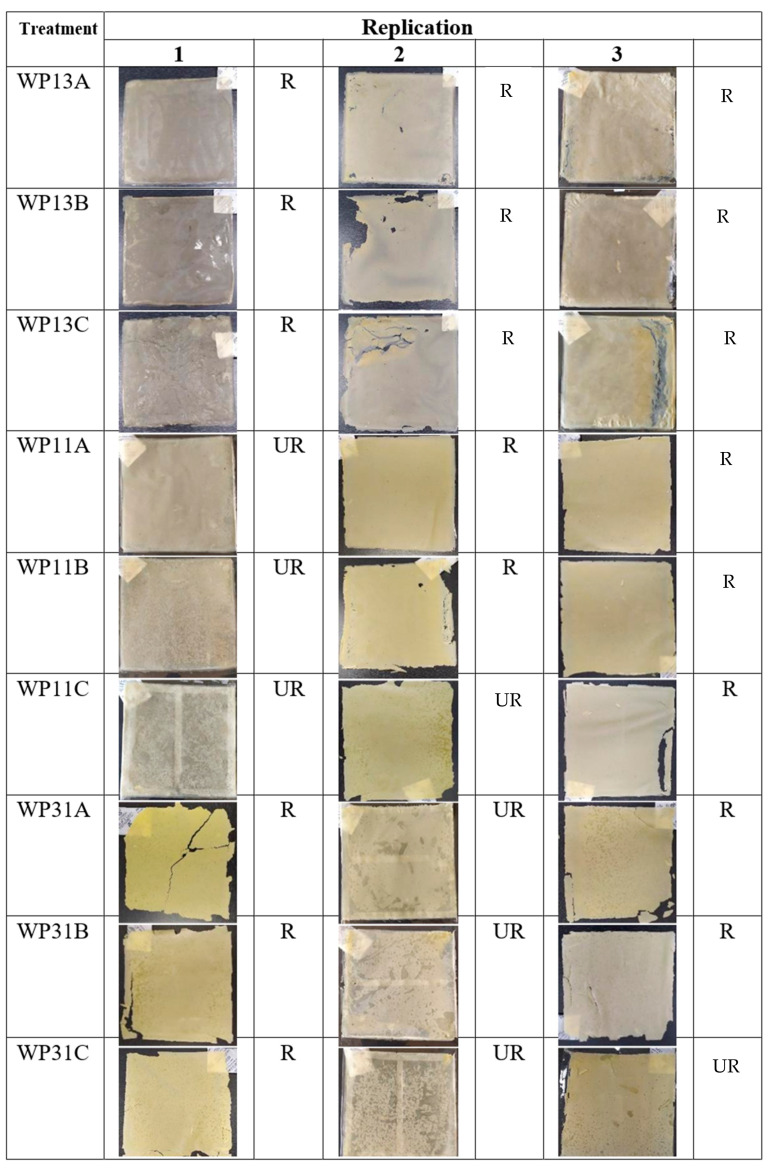
The appearance of antimicrobial composite edible film. R = removable; UR = unremovable.

**Figure 2 membranes-12-00636-f002:**
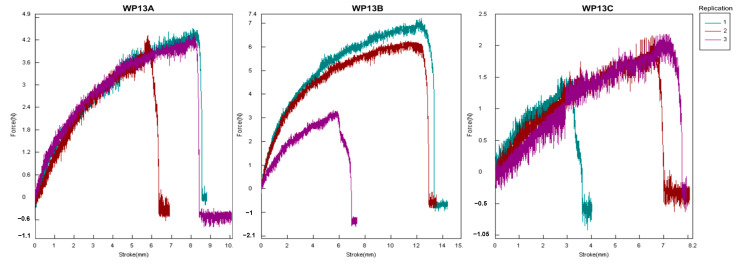
The tensile strength curve of antimicrobial composite edible film.

**Table 1 membranes-12-00636-t001:** Treatment in the making of composite antimicrobial edible film.

		Whey: Starch Ratio (*v*/*v*)		
		1:3	1:1	3:1
Glycerol Ratio (*v*/*v*)	20%	WP13A	WP11A	WP31A
	33%	WP13B	WP11B	WP31B
	45%	WP13C	WP11C	WP31C

**Table 2 membranes-12-00636-t002:** The treatment of composite antimicrobial edible film formulation.

		Whey Volume (mL)	Whey Total Solid (g)	Starch Volume (mL)	Starch Total Solid (g)	Glycerol Volume (mL)
WP13	A	50	2.5	150	7.5	1
	B	50	2.5	150	7.5	2.5
	C	50	2.5	150	7.5	4
WP11	A	100	5	100	5	1
	B	100	5	100	5	2.5
	C	100	5	100	5	4
WP31	A	150	7.5	50	2.5	1
	B	150	7.5	50	2.5	2.5
	C	150	7.5	50	2.5	4

**Table 3 membranes-12-00636-t003:** Thickness, tensile strength, and elongation at break.

Treatment	Thickness (mm)	Tensile Strength (N/mm^2^)	Elongation at Break (%)
WP13A	0.23 ± 0.01	0.97 ± 0.04	13.86 ± 2.36
WP13B	0.21 ± 0.00	0.81 ± 0.29	19.62 ± 7.20
WP13C	0.26 ± 0.01	0.22 ± 0.04	10.56 ± 4.53

**Table 4 membranes-12-00636-t004:** Water vapour permeability.

Treatment	WVP (RH 100%–35%) (10^−10^·g/m·s·Pa)	WVP (RH 75%–35%) (10^−10^·g/m·s·Pa)
WP13A	3.56 ± 1.28	12.18 ± 2.04
WP13B	3.41 ± 1.13	9.84 ± 1.50
WP13C	4.84 ± 1.80	14.59 ± 1.43

**Table 5 membranes-12-00636-t005:** L * a * b result of colour.

Treatment	L *	a *	b *
WP13A	94.21 ± 2.30	1.64 ± 0.17	9.65 ± 0.76
WP13B	96.15 ± 0.57	1.43 ± 0.19	8.55 ± 1.00
WP13C	94.27 ± 0.76	1.58 ± 0.23	9.51 ± 0.65

**Table 6 membranes-12-00636-t006:** The result of antimicrobial activities.

Treatment	Clear Zone (mm)			
	*P. aeruginosa*	*S. aureus*	*S. thypimurium*	*E. coli*
Whey	5.00 ± 0.00	5.00 ± 0.00	5.00 ± 0.00	5.00 ± 0.00
Unfermented whey	5.00 ± 0.00	5.00 ± 0.00	5.00 ± 0.00	5.00 ± 0.00
Fermented whey	18.50 ± 8.73	5.00 ± 0.00	5.00 ± 0.00	5.00 ± 0.00
WP13A solution	5.00 ± 0.00	5.00 ± 0.00	5.00 ± 0.00	5.00 ± 0.00
WP13B solution	5.11 ± 0.71	5.00 ± 0.00	5.00 ± 0.00	5.00 ± 0.00
WP13C solution	7.00 ± 1.73	5.00 ± 0.00	5.00 ± 0.00	5.00 ± 0.00

## Data Availability

Data is contained within the article.

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
