# Peer review of "Characterization of Antimicrobial Composite Edible Film Formulated from Fermented Cheese Whey and Cassava Peel Starch"

_membranes, 2022, doi:10.3390/membranes12060636_

Round 1

Author Response

Dear Reviewer,

Thank you very much for your valuable input and recommendation. We do our best to revised the document, however there are some limitation and we hope you can accept it. Please kindly check the attachment below.

Best Regards,
Authors

Reviewer 2 Report

The authors need to cover well previous research on cassava starch /  cheese whey composites in the introduction and also in discussion parts so that the innovation of the submitted manuscript will be obvious. Here are just an example that is not covered:

Edible Films of Whey and Cassava Starch: Physical, Thermal, and Microstructural Characterization. Coatings 2020, 10, 1059; doi:10.3390/coatings10111059

The tensile strength of the composites is extremely low (below 1 MPa). Could the authors explain how such composites will be used? It will be too difficult to handle; if the humidity is relatively high, there will be no strength. What about the tensile modulus of these composites? Please add stress/strain curves for the tensile test.

Author Response

Dear Reviewer,

Thank you very much for the valuable input and recommendation. Please kindly check attached file below.

Best Regards,

Authors

Reviewer 3 Report

I suggest that  this reseach can be accepted now.

Author Response

Dear Reviewer,

Thank you very much for your valuable input and suggestion.

Best Regards,

Authors

Round 2

Reviewer 2 Report

The authors reponded reasonably to the reviewers' comments.

This manuscript is a resubmission of an earlier submission. The following is a list of the peer review reports and author responses from that submission.

Round 1

Reviewer 1 Report

This manuscript reports formulation and investigations of the antimicrobial, physical, mechanical, and barrier properties of the fermented cheese whey/cassava peel starch edible films. The manuscript contains novelty but needs major improvement before it can be considered for publication.

General comments:

  1. This manuscript needs to be sent to the English editing service. There are many grammatical and spelling errors.

Abstract

  1. Line 16 – What does it mean by 75% starch and why choose this value
  2. Line 18 – The manuscript did not only investigate physical properties but also mechanical and water vapor permeability properties. Please consider including these in the Abstract and Introduction sections.
  3. Line 24 – Specify abbreviation for C. tropicalis
  4. Line 25 – Specify WP13B formulation in detail instead of just giving the abbreviation

Introduction

  1. Consider discussing the gap in the literature that the work is trying to fill in
  2. Consider specifying the novelty of the work

Materials and Methods

  1. Include brand, country, and manufacturer for all chemicals and equipment used for the work
  2. Table 2 – Why the unit used for glycerol is in mL (volume) and not weight since Table 1 tabulates glycerol ratio (w/w)?
  3. Some of the chemicals/raw materials and equipment were not listed on page 4. E.g. yeast mold agar, chloramphenicol, etc. Please check all in Sections 2.1, 2.2, 2.3, 2.4, and 3.5. Include brand, country, and manufacturer for all chemicals and equipment used for the work.
  4. Line 96 – [20] Please check the right format to cite in the manuscript text
  5. Please detail out the equipment for certain processes. E.g. incubated using.. [Line 94], pasteurized using.. [Line 98], etc. Please check the whole Section 2. Materials and Methods
  6. Section 3.5, Antimicrobial Activity Identification should be 2.5
  7. Line 126 – Please specify abbreviations for all bacteria names when they are first mentioned in the manuscript
  8. Consider including other antimicrobial analysis instead of just disc diffusion analysis to support the antimicrobial results.

Results

  1. Line 132 – WP13A, B, or C?
  2. Please include discussion on WP11
  3. Line 146 – Do you want to produce thin or thick films? Explain why
  4. Lines 147-148 – Is this means that WP13B contains the highest amylose content? It is suggested for the authors to determine the amylose content for each film to support the statement.
  5. Explain why obtained the trends of the results for the physical, mechanical, WVP, and antimicrobial analysis
  6. Explain why the lowest WVP for WP13B films?
  7. Base on Table 4, which sample films exhibit the best physical properties?
  8. Base on Table 4, why was no clear zone/antimicrobial activity observed against S. aureus, S. thypimurium, and E. coli. Why P. aeruginosa only? Why higher for WP13C compared to A and B? Please explain.

Discussion

  1. Be specific on which discussion correspond to which results.
  2. It is suggested to combine results and discussion in one section instead of having separate sections for clear understanding of the readers.
  3. The discussion can be further improved by providing more detailed discussions on the results obtained. Some of the discussion on tabulated results are missing.

References

  1. The format of the references needs to be checked. Ensure consistency.

Author Response

Dear Reviewer, we are grateful for your appreciation, consideration, valuable suggestions, and recommendations for the further quality improvement of our manuscript. We have taken your suggestion for further critical assessment in the manuscript very seriously. We have added all needful material wherever important corrections were necessary throughout the manuscript in order to perform corrections in a highly critical manner. All the necessary actions have been performed in the revised manuscript by using track change mode.

Reviewer 2 Report

The article entitled " Characterization of Antimicrobial Composite Edible Film Formulated from Fermented Cheese Whey and Cassava Peel Starch" showed the impacts of whey on physical properties and antimicrobial activities of edible films. The article is interesting, the experiments are accurately described. I suggest this article can be accepted after some modifications. 1.line 52: More details about the antimicrobial peptides should be provided, such as variety and molecular weight.

2.line 186: Is it possible that the occurrence of Maillard reaction led to the changes of color?

3.line 195: Is there any way to avoid the high temperature treatment, since it can result in the denaturation of proteins.

4.line 204: Is this film suitable for meat or fruits packaging?

Author Response

(The authors gave the same response as above.)

Round 2

Reviewer 1 Report

This manuscript reports formulation and investigations of the antimicrobial, physical, mechanical, and barrier properties of the fermented cheese whey/cassava peel starch edible films. The manuscript contains novelty but still needs major improvement after the first correction before the manuscript can be accepted for publication.

The authors have made several improvements based on the previous comments, but many still have not been improved.

General comments:

  1. This manuscript needs to be sent to the professional English editing service. There are still many grammatical and spelling errors.
  2. Please check the right format to cite the references in the WHOLE MANUSCRIPT text.
  3. Please specify abbreviations for all bacteria names when they are FIRST MENTIONED in the manuscript and then specify the abbreviations in the brackets.

Results and discussion:

  1. Results and discussion should be combined in one section instead of having separate sections for a clear understanding of the readers. I don’t think it is a problem with the journal format.
  2. The results and discussion are still clearly lacking in explanation. They shall be further expanded and improved by providing more detailed discussions on the results obtained. Some of the discussion on tabulated results are missing. For example, the authors should explain why they obtained the trends of the results for the physical, mechanical, WVP, and antimicrobial analyses.

Conclusions

  1. Conclusions should be amended according to the results and discussion section.

Thank you.

Author Response

Dear Reviewer, we are very sorry fo the late respons because we have to finish the proof read session that maybe still leaves little error in the gramatical or english. We are very grateful for the further quality improvement of our manuscript. We All the necessary actions have been performed in the revised manuscript by using track change mode.

General comments:

  • This manuscript needs to be sent to the professional English editing service. There are still many grammatical and spelling errors.

- The english proof reading already done 

  • Please check the right format to cite the references in the WHOLE MANUSCRIPT text.

- All the necessary actions have been performed in the revised manuscript 

  • Please specify abbreviations for all bacteria names when they are FIRST MENTIONED in the manuscript and then specify the abbreviations in the brackets.

- All the necessary actions have been performed in the revised manuscript, line 156-157

Results and discussion:

  • Results and discussion should be combined in one section instead of having separate sections for a clear understanding of the readers. I don’t think it is a problem with the journal format.

- All the necessary actions have been performed in the revised manuscript, line 159-273

  • The results and discussion are still clearly lacking in explanation. They shall be further expanded and improved by providing more detailed discussions on the results obtained. Some of the discussion on tabulated results are missing. For example, the authors should explain why they obtained the trends of the results for the physical, mechanical, WVP, and antimicrobial analyses.

- All the necessary actions have been performed in the revised manuscript, line 159-273

Conclusions

  • Conclusions should be amended according to the results and discussion section.

- All the necessary actions have been performed in the revised manuscript, line 278-288